# Annotations Mitigate Post-Training Mode Collapse

**Jacob Mitchell Springer** [†1]  **Madhu Advani** [2]  **Lukas Aichberger** [†3]  **Arwen Bradley** [2]  **Eran Malach** [2]
**Omid Saremi** [2]  **Sinead Williamson** [2]  **Preetum Nakkiran** [2]  **Etai Littwin** [2]  **Aditi Raghunathan** [1]

## Abstract

Post-training (via supervised fine-tuning) improves instruction-following, but often induces *semantic mode collapse* by biasing models toward low-entropy fine-tuning data at the expense of the high-entropy pretraining distribution. Crucially, we find this trade-off worsens with scale. To close this semantic diversity gap, we propose *annotation-anchored training*, a principled method that enables models to adopt the preference-following behaviors of post-training without sacrificing the inherent diversity of pretraining. Our approach is simple: we pretrain on documents paired with semantic annotations, inducing a rich annotation distribution that reflects the full breadth of pretraining data, and we preserve this distribution during post-training. This lets us sample diverse annotations at inference time and use them as anchors to guide generation, effectively transferring pretraining's semantic richness into post-trained models. We find that models trained with annotation-anchored training can attain 6× less diversity collapse than models trained with SFT, and improve with scale.

## 1. Introduction

Pretraining on web-scale corpora exposes language models to an enormous range of topics, genres, entities, and writing styles. Even modest-sized models have seen more tokens during pretraining than any human could read in many lifetimes, capturing patterns from across the open web. This breadth—spanning topics, registers, languages, and disciplines—is central to what makes pretrained language models broadly useful across the diverse downstream tasks where they are eventually deployed. Deploying these models in a user-facing setting typically requires an additional *post-training* step that adapts a base model into one that follows instructions and replies helpfully. In this paper, we place our focus on the instruction-tuning step, where the model is trained with supervised fine-tuning (SFT) on instruction data to improve helpfulness, formatting, safety, and instruction-following (Ouyang et al., 2022). A growing body of evidence suggests that this type of post-training can inadvertently compress the space of plausible outputs, yielding models that are high-quality yet *semantically homogeneous* (Kirk et al., 2023; Hamilton, 2024; Murthy et al., 2025; Yun et al., 2025; O'Mahony et al., 2024; Zhang et al., 2025b; Jiang et al., 2025).

One might expect larger models—with broader world knowledge and richer representations—to better preserve semantic diversity under post-training. Zhang et al. (2025b) first reports the opposite trend: instruction tuning can induce semantic mode collapse that *worsens with model size*. We observe the same pattern across model families and sampling methods (Figure 3). This is not because base models are less diverse: before post-training, diversity increases with model size. We observe this growing semantic diversity gap only after post-training.

We propose that this failure is rooted in what instruction-tuning optimizes. SFT is typically framed as "make the model match the post-training data distribution." However, the post-training distribution contains both changes that we want, such as improved conditional responses given a semantic intent, and changes we may not want, such as inheriting the low-entropy response semantics of the (often limited) post-training data. To separate these, we factor post-training into: *(i) a semantics-conditional response distribution* (what post-training should change) and *(ii) a semantic distribution* (what post-training should often preserve from pretraining).

We introduce *annotation-anchored training*, which preserves the high-entropy semantic diversity learned during pretraining while still incorporating the improvements that arise from instruction tuning (Figure 1). Concretely, our method has three components. First, during pretraining, we augment pretraining documents with semantic annotations so the model learns a rich and diverse annotation distribution. Second, during post-training, we train the model to

---

[†]Work done in part while interning at Apple. [1]Carnegie Mellon University [2]Apple [3]Johannes Kepler University Linz. Correspondence to: Jacob Mitchell Springer <jspringer@cmu.edu>.

*Proceedings of the 43rd International Conference on Machine Learning*, Seoul, South Korea. PMLR 306, 2026. Copyright 2026 by the author(s).

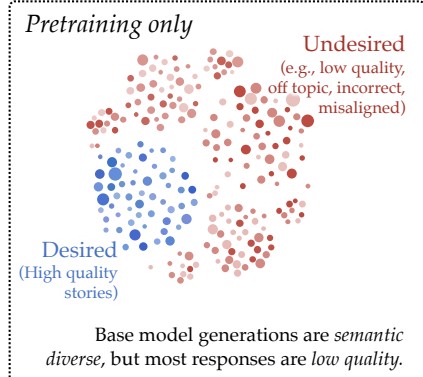 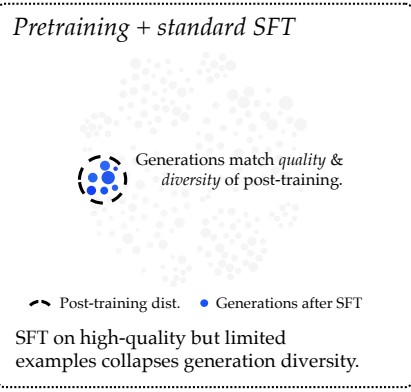 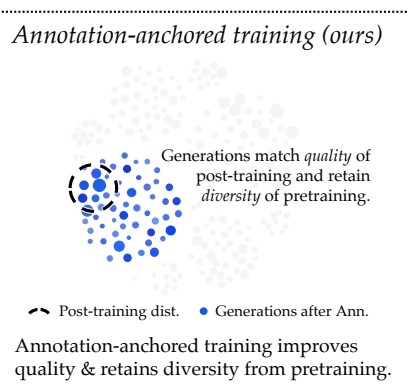

*Figure 1.* Annotation-anchored training preserves the semantic diversity of pretraining while adopting the quality of post-training. *(Left)* Base models produce semantically diverse outputs, but many generations are low quality or off topic. *(Middle)* Standard SFT concentrates generations on the high-quality but narrow post-training distribution, collapsing semantic diversity. *(Right)* Annotation-anchored training matches the quality of post-training while retaining the semantic diversity of pretraining.

generate responses conditioned on the provided annotations, while preventing gradients from updating the annotation prediction behavior, so the semantic distribution remains anchored to its pretrained state. Finally, at inference time, the model first samples an annotation (a semantic plan) and then generates a response conditioned on that plan. This simple change decouples the semantic distribution from the post-training data and mitigates semantic mode collapse. See Figure 2 for a schematic of the method.

The authors declare no competing interests relevant to this work.

## 2. Related Work

**Diversity, novelty, and mode collapse in language models.** A growing line of work argues that modern LMs can be *high-quality yet semantically homogeneous*, and that this homogeneity is not well captured by surface-form diversity. NoveltyBench explicitly evaluates whether models can produce *distinct but valid* outputs and documents semantic collapse that is often strongest in post-trained models (Zhang et al., 2025b). Complementary evidence appears in studies of narrative "persona" collapse (Hamilton, 2024) and population-level reductions in viewpoint or conceptual coverage after alignment (Murthy et al., 2025). At a broader behavioral level, aligned models can become less random/creative than their base counterparts (West & Potts, 2025), and even human writing assisted by LMs can exhibit reduced content diversity (Padmakumar & He, 2023). Recent benchmark-driven efforts also emphasize that "diversity" spans linguistic, semantic, and discourse dimensions, and that comparisons can flip depending on the metric and sampling regime (Guo et al., 2025).

**Measuring diversity beyond surface variation.** Because naive $n$-gram diversity can be gamed, recent work stresses

semantic-aware evaluation (Tevet & Berant, 2021; Guo et al., 2025), distributional metrics such as MAUVE (Pillutla et al., 2021), and embedding-based similarity (Zhang et al., 2019; Zhu et al., 2018; Caccia et al., 2018). Our semantic-entropy framing aligns with this shift.

**Post-training, alignment, and distributional shift.** Standard instruction-tuning and alignment pipelines (SFT + preference learning / RLHF) are now the default for deployment (Ouyang et al., 2022; Christiano et al., 2017; Ziegler et al., 2019; Stiennon et al., 2020; Bai et al., 2022; Rafailov et al., 2023). Multiple empirical studies show that these stages can systematically reduce per-input diversity even as they improve judged helpfulness or robustness (Kirk et al., 2023; O'Mahony et al., 2024). The "format" and stylistic conventions learned during post-training can further concentrate outputs into a narrow response manifold (Yun et al., 2025). Our work is motivated by the same phenomenon, but targets a more specific causal hypothesis: post-training transfers an *unwanted low-entropy semantic prior* from the post-training dataset, even when the desired change is primarily in the semantics-*conditional* response behavior.

**Training-time objectives for preserving or encouraging diversity.** A classic approach is to alter the learning objective so likelihood training does not over-concentrate probability mass. Early dialogue work promotes diversity via mutual-information-inspired objectives (Li et al., 2016), while confidence/entropy regularization can discourage overly peaked predictive distributions (Pereyra et al., 2017). Unlikelihood training directly penalizes degenerate repetitions and has become a standard tool for combating collapse-like behaviors under likelihood objectives (Welleck et al., 2019). More recent post-training-specific methods explicitly aim to preserve diversity during SFT, e.g., via game-theoretic / entropy-regularized formulations (Li et al., 2024) or by encouraging more diffuse output distributions

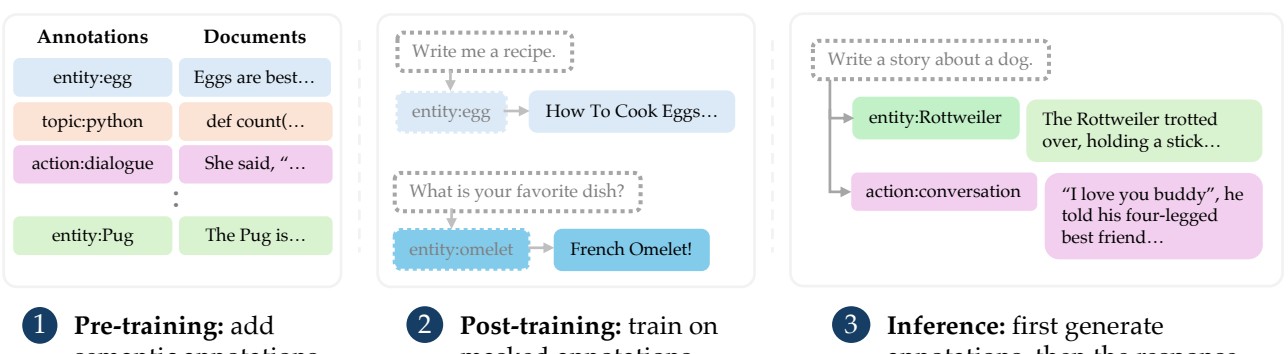

**1** **Pre-training:** add semantic annotations.

**2** **Post-training:** train on masked annotations.

**3** **Inference:** first generate annotations, then the response.

*Figure 2.* SFT can induce *semantic mode collapse*, leading to models with limited generation diversity. Our method, *annotation-anchored training*, mitigates this by making semantics explicit as natural-language annotations: during pretraining, the model learns a rich distribution over annotations; during post-training, we mask the loss of the annotations, preserving the annotation distribution from pretraining. At inference time, we can thus sample annotations that capture the diversity of the pretraining distribution to guide the final generation.

during training (Zhang et al., 2024). Our anchored objective is closest in spirit to these training-time interventions, but differs in what is being preserved: rather than regularizing the response distribution in aggregate, we explicitly preserve the *semantic marginal* by mediating generation through an explicit semantic variable. Methods like entropy regularization or KL-constrained fine-tuning penalize deviation from a reference distribution but do not distinguish *which* distributional properties to preserve; our anchored objective is complementary in that it explicitly preserves the semantic marginal while permitting arbitrary changes to the conditional.

**Inference-time decoding and sampling for diverse generation.** A large literature improves diversity at inference: diverse beam search (Vijayakumar et al., 2016), stochastic methods (Kool et al., 2019), nucleus/top-$p$/top-$k$ sampling (Holtzman et al., 2019; Fan et al., 2018), typical sampling (Meister et al., 2023), contrastive decoding (Su et al., 2022), and higher-level interventions such as arithmetic and verbalized sampling (Vilnis et al., 2023; Zhang et al., 2025a). These methods are complementary: they shift the diversity–quality operating point but do not address the training-time transfer of a low-entropy semantic prior into the model.

**Controllable generation, planning variables, and latent structure.** Many methods introduce explicit control or latent variables to separate *what to say* from *how to say it*: control codes (Keskar et al., 2019), gradient-based steering (Dathathri et al., 2019), and explicit planning for long-form generation (Yao et al., 2019). In dialogue, latent-variable models (often CVAEs) encode multiple plausible discourse intents to increase conditional diversity (Zhao et al., 2017; Cao & Clark, 2017). Recent systems also combine base and aligned models to trade off diversity and instruction-following at inference (Wang et al., 2025). Our approach fits squarely in this tradition—introducing an explicit semantic

mediator—but with a distinct goal: we use annotations to *preserve a pretraining-learned semantic prior through post-training*, rather than enforcing a user-specified attribute or post-hoc steering signal.

## 3. Tracking Semantic Collapse

In this section, we quantify how post-training changes the semantic diversity of model generations.

### 3.1. Experimental setup

**Model families.** We consider two model families: the Qwen2.5 series (Yang et al., 2025), spanning 0.5B, 3B, 14B, and 72B parameters, and the Llama 3 series (Grattafiori et al., 2024), spanning 1B, 3B, 8B, and 70B parameters. For each model size we pair the publicly released base model with its officially released instruction-tuned counterpart, which keeps the architecture and pretraining data fixed across the base-vs. post-trained comparison.

**Semantic diversity evaluation.** We benchmark with STO-RIES, prompting models to generate a story beginning with "Once upon a time" (base models use prefilling). We map generations $y \sim f(\cdot \mid x)$ to semantic labels via an annotation function $a \colon y \mapsto z$ and compute the semantic entropy of the pushforward distribution (Kuhn et al., 2023). An LLM judge (Qwen3-30B-A3B-Instruct) assigns single-word labels along eight attributes (SUMMARY, MAIN CHARACTER NAME, LOCATION, GENRE, NARRATIVE STRUCTURE, CULTURAL CONTEXT, WORLDBUILDING FLAVOR, IMAGERY); we report mean entropy across these attributes (see Appendices E.1 and E.2). Higher entropy on a given attribute corresponds to greater across-sample variation in that aspect of the story—for instance, high LOCATION entropy means stories are set in more varied places.

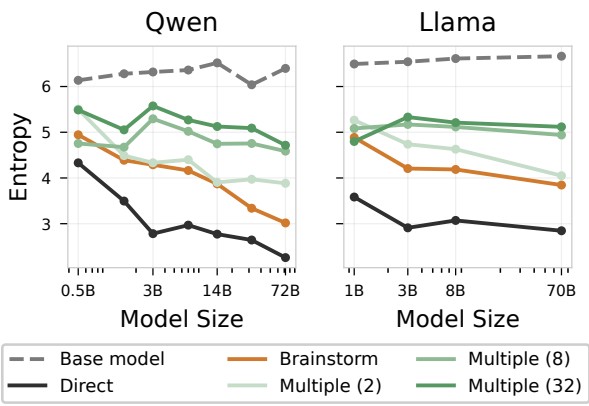

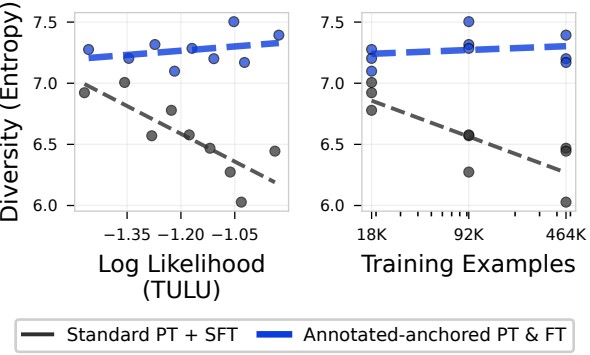

*Figure 4.* Semantic diversity (entropy) as a function of the log-likelihood of the post-training validation data for different model sizes and post-training hyperparameters *(left)*, and as a function of the number of post-training examples *(right)*. Standard SFT exhibits a negative correlation between likelihood and diversity, and between post-training dataset size and diversity. Our method, *annotation-anchored training*, reverses both relationships.

*Figure 3.* Semantic diversity (entropy) as a function of model size across model families on the STORIES benchmark. The instruction-tuned models exhibit inverse scaling of diversity with respect to model size across sampling methods, with the notable exception of *multiple* sampling for Llama, which remains roughly constant. All prompting methods reveal a substantial drop in the semantic diversity of the instruction-tuned models compared to the base models.

**Sampling.** We evaluate three sampling procedures for instruction-tuned models: *direct* (standard prompting); *brainstorm* (sample ideas, then condition the response on them) (Yao et al., 2019; Ahmed et al., 2025); and *multiple* (*n*) (sample *n* responses in a shared context with a diversity instruction). These procedures span a range of strategies, from standard single-response prompting to explicit multi-sample generation, and let us test whether the diversity drop observed under direct prompting persists when models are given more freedom to produce varied outputs. See Appendix D.2 for details.

### 3.2. Results

**Larger base models are not less diverse, but larger post-trained models are.** Larger Qwen and Llama base models exhibit *greater* semantic diversity (Figure 3, dashed), consistent with the intuition that scale supports richer generative coverage. In contrast, the semantic diversity of their post-trained counterparts *declines* with scale (Figure 3, solid), revealing an inverse scaling trend that emerges only after post-training. This phenomenon was first documented by Zhang et al. (2025b) for direct prompting in instruction-tuned models; our results broaden their finding by showing that it persists under prompting strategies designed to elicit diversity. Indeed, while alternative prompting—especially sampling multiple responses in a shared context—raises semantic entropy, it does not close the gap between base and post-trained models, nor does it remove the residual inverse scaling with respect to model size. This rules out the possibility that the observed collapse is purely an artifact of single-response decoding and would disappear with the

right prompting recipe.

**Fitting post-training data correlates with reduced diversity.** We next examine whether semantic collapse tracks how closely models match the post-training distribution by comparing post-training validation log-likelihood and generation diversity across hyperparameters, training duration, and dataset size. We consistently find that models with lower post-training loss (higher likelihood) generate less diverse outputs (Figure 4, left), consistent with the intuition that when post-training data has reduced semantic entropy, likelihood optimization concentrates probability mass onto the fewer semantic modes that the dataset rewards. This helps explain why larger models—able to fit the post-training distribution more tightly than smaller ones—exhibit decreased semantic diversity, and it also yields a counterintuitive prediction that we confirm empirically: increasing the number of post-training examples further degrades semantic diversity (Figure 4, right). Both predictions hold uniformly across the model sizes and training durations we evaluated, suggesting that the relationship between dataset fit and semantic collapse is a robust phenomenon rather than an artifact of any single hyperparameter setting or evaluation choice.

**A distributional-transfer problem.** Taken together, these observations suggest that the post-training dataset implicitly specifies a low-entropy semantic prior over what models should generate, and that standard likelihood training transfers that narrow prior directly into the resulting model. The crux of the issue, then, is that the desirable high-entropy semantic prior of the pretraining distribution *fails to transfer after post-training*, and is in fact actively suppressed whenever the post-training data has a tighter semantic distribution than what the base model already represents. In Section 4, we formalize this view as a factorization of the response

distribution and propose a training objective that explicitly preserves the high-entropy semantics learned during pre-training, while still allowing post-training to freely update the conditional response behavior in the ways we want.

## 4. Annotation-Anchored Training

This section introduces *annotation-anchored training*, a distributional view that separates what post-training should *change* from what it should *preserve* from pretraining. We propose that augmenting the training data with natural-language *annotations* and preserving their distribution during post-training transfers the semantic diversity of pretraining into the resulting post-trained models, mitigating the collapse induced by standard SFT.

### 4.1. Preserving distributional aspects from pretraining

Our goal is to propose a fine-tuning method that preserves the semantic diversity learned during pretraining while still adopting the semantics-conditioned behavioral improvements that come from post-training. Throughout this section, we write $x$ for a context (e.g., a prompt or instruction), $y$ for a response (a document, continuation, or completion), and $z$ for a semantic variable that captures the topic, intent, or other high-level structure of the response.

**A factorized view of pretraining and post-training.** We assume the pretraining distribution admits a semantic factorization,

$$P(y) = \int R(z) \, Q(y \mid z) \, dz, \qquad (1)$$

in which $R(z)$ is a high-entropy semantics distribution over the latent modes spanned by the pretraining corpus—topics, intents, narrative structures, and so on—while $Q(y \mid z)$ is the textual-form distribution that generates surface text $y$ conditioned on those semantics. This factorization separates *which* mode is being expressed from *how* the response is rendered as text, and lets us reason about each part of the distribution independently.

The post-training distribution can be factored similarly:

$$P^\star(y \mid x) = \int R^\star(z \mid x) \, Q^\star(y \mid x, z) \, dz. \qquad (2)$$

$Q^\star(y \mid x, z)$ captures *how* to respond to a prompt $x$ given an intent/semantic mode $z$, while $R^\star(z \mid x)$ captures *which* semantic modes are represented by post-training conditioned on the prompt $x$. When the post-training semantics $R^\star(\cdot \mid x)$ has substantially smaller entropy than the pretraining semantics $R(\cdot \mid x)$, matching $P^\star$ (via post-training) will induce semantic mode collapse.

**Anchoring semantics while adopting post-training conditional behavior.** To decouple the semantic marginal from the (desirable) conditional changes of post-training, we propose that post-training should train the model to match the following distribution:

$$P_R^\star(y \mid x) = \int R(z \mid x) \, Q^\star(y \mid x, z) \, dz. \qquad (3)$$

Intuitively, $P_R^\star$ keeps the conditional response behavior of post-training $Q^\star(y \mid x, z)$ but preserves the semantics of pretraining $R(z \mid x)$.

### 4.2. Annotation-anchored training

To train a model to match the conditional response behavior of post-training while preserving the semantic distribution of pretraining, we propose the *annotation-anchored training* pipeline, in which we make explicit the semantic variable $z$ by annotating the semantics of each document. Our method is defined as follows (see Figure 2 for an illustration):

---

**Annotation-Anchored Training (definition)**

**During pretraining:** Given a document $y$ and a textual representation $z$ of the document's semantics, train autoregressively on the concatenated sequence $z \parallel y$, where $\parallel$ denotes string concatenation and the semantic annotations have been prepended to the document.

**During post-training:** Given a prompt $x$, a response $y$, and the semantic annotations $z$ of the response, train autoregressively on the concatenated sequence $x \parallel z \parallel y$. To prevent post-training from updating the pretrained conditional distribution $R(z \mid x)$, mask out the loss of both the prompt $x$ and the semantic annotation $z$ during the backpropagation of the gradient. By masking $z$, the model will preserve its pretrained (high-entropy) initialization.

**At inference:** Given a novel prompt $x'$, sample the model autoregressively as usual. The model will implicitly generate annotations $z'$ conditioned on the prompt, and subsequently a response $y'$ conditioned on both the prompt and the annotations.

---

### 4.3. Practical considerations

The *annotation-anchored training* leaves two practical questions: (i) how should the semantic variable $z$ be represented, and (ii) how should we induce a high-entropy yet coherent conditional distribution $z \mid x$ during pretraining.

| Text | Annotation |
|------|-----------|
| Just as the work was well under way, Mrs. Sinclair informed the Britts that she and Whyn must leave for the city. She had her work to do there without which they could not live... | TOPIC: family relationships DOMAIN: literature ACTION: departure ENTITIES: Mrs. Sinclair; Whyn; Britts |

*Table 1.* Example (partial) pretraining document with annotations.

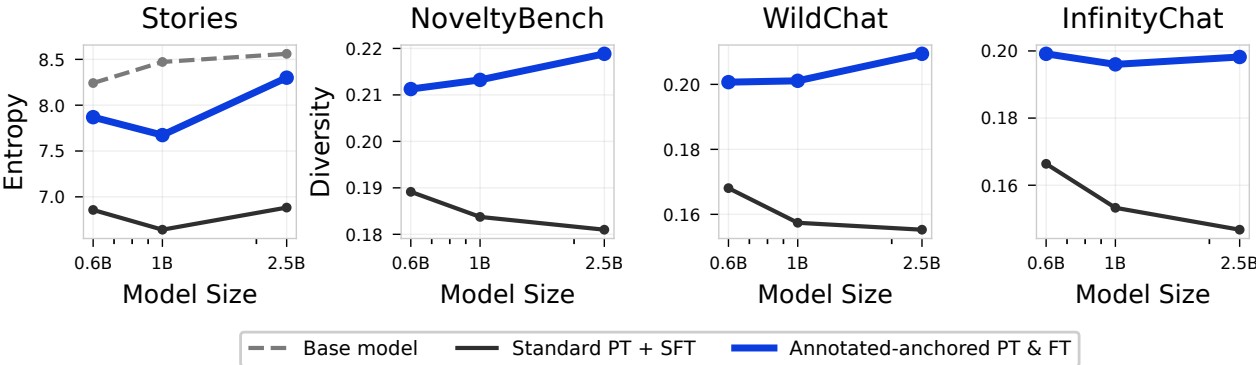

*Figure 5.* Comparing the semantic diversity of standard and annotation-anchored models across scales on the STORIES, NOVELTYBENCH, WILDCHAT, and INFINITYCHAT benchmarks. Annotation-anchored models maintain higher diversity than standard models across all benchmarks and model scales. For STORIES, the 2.5B-parameter annotation-anchored model closes the semantic diversity gap with the base model by roughly 85%.

**Choice of $z$: concise semantic annotations.** We instantiate $z$ as a short list of structured tags (key-value pairs) that summarize salient semantic attributes of text (entities, actions, setting, domain, location, and others; see Table 1 for an example and Appendix B.4 for the full schema). Annotations are represented directly as text and thus are natively compatible with standard language models. Because they are derived from diverse pretraining corpora, they induce a rich semantic prior. In practice, we generate annotations with a language model (see Appendix B.5 for compute requirements).

**Inducing $z \mid x$ during pretraining via chunked annotations.** We annotate documents at the chunk level to make generated annotations both diverse and locally relevant. Given a document split into contiguous chunks $x_1, \ldots, x_n$, we annotate each chunk to produce $z_1, \ldots, z_n$. We then train autoregressively on the interleaved sequence:

$$\langle z_1 \rangle \, x_1 \, \langle z_2 \rangle \, x_2 \, \cdots \, \langle z_n \rangle \, x_n.$$

This ordering teaches the model to condition annotations on prior context (e.g., a prompt), and the resulting conditional distribution serves as a high-quality prior for the instruction-tuning format (Section 5.1).

**Post-training with anchored annotations.** To obtain annotations for post-training examples, we apply the same annotation scheme used during pretraining to each target response in the post-training dataset.

**Principles for choosing annotations.** Useful annotations describe the choices we want the model to vary across samples but that are not already determined by the prompt. For the creative tasks we study, we instantiate this with concise tags for salient semantic choices: topic, entities, actions, setting, domain, and location. This criterion can extend beyond creative tasks, for instance to mathematics: when the

right approach is uncertain, annotations could describe the intended proof strategy, key idea, or representation, rather than a particular final answer. We leave such extensions to future work.

# 5. Experiments

We systematically evaluate *annotation-anchored training* against standard SFT post-training.

## 5.1. Experimental setup

**Annotations.** Annotations describe semantic content (topic, domain, sentiment, entities, locations, etc.) rather than surface form. We use *Qwen3-30B-A3B-Instruct* (Yang et al., 2025) to extract salient attributes as variable-length `<key>:<value>` tags. The approach does not require perfect annotations: it preserves the *entropy* of the learned distribution rather than per-example correctness, so occasional annotator noise does not degrade training. See Table 1 and Appendix B for details.

**Benchmarks and metrics.** We evaluate on STORIES (Section 3) and three diversity-oriented dialog benchmarks (NOVELTYBENCH, WILDCHAT, INFINITYCHAT) (Zhang et al., 2025b; Zhao et al., 2024; Jiang et al., 2025). For Stories, we use *semantic entropy* (Section 3); for dialog, mean pairwise cosine dissimilarity of Qwen3-Embedding-0.6B embeddings (Zhang et al., 2025b; Yang et al., 2025). Quality is judged by Qwen3-30B-A3B-Instruct (Appendices E.1 and E.3).

**Pretraining.** We pretrain 0.6B, 1B, and 2.5B models on Dolma/Dolmino data (Soldaini et al., 2024) (web, books, Wikipedia, forums, math) using OLMo-2 (OLMo et al., 2024), training each for the Chinchilla-optimal token count (12B, 20B, 50B respectively) (Hoffmann et al., 2022). All

| Prompt | Standard SFT | Annotation-anchored |
|---|---|---|
| Write a story that begins with "Once upon a time". | *Once upon a time in a quaint little village nestled between rolling hills and whispering forests, there was a young girl named Alice who had a heart as vast as the ocean. Her days were filled with laughter, the moonlight shining down upon her tiny cottage, and the world seemed full of wonder and adventure...* | TOPIC:world-building    TOPIC:global issues ... 
 *Once upon a time in a world vastly different from the one we know, a startling change was in the air—the crime rate had soared, followed by a sudden, alarming decline in the medical field. Something similar happened to the geographical context: the cities grew strange, and humans became less relatable to one another...* |
| | *Once upon a time there was a young girl named Lily who lived nestled in a patch of vibrant flowers. Her father was a gardener preparing for the upcoming underground art festival, which promised to be a magical experience within their new paradise...* | LOCATION:Baghdad    ENTITY:Mir Hussein ... 
 *Once upon a time in a quiet corner of the world where time seemed to bend to the rhythm of familiar human emotions, a young woman named Amira found herself drawn to a shadow, a person whose silhouette seemed to dance with the world only to move apart. Their names were Mir Hussein and Abbey...* |

*Table 2.* Representative generations illustrating qualitative differences between standard SFT and annotation-anchored training.

variants match total tokens and FLOPs; in annotated pretraining, annotations *replace* content tokens. *Standard* models follow standard autoregressive pretraining; *Annotated* models train on documents split at double newlines and annotated per chunk, teaching a high-entropy conditional distribution over annotations that serves as the semantic anchor. See Appendix A for details.

**Post-training.** Each model is post-trained on TULU3 SFT for one epoch (Lambert et al., 2024; Song et al., 2024); see Appendix C.1. Standard models use standard SFT; annotation-anchored models annotate each response and train on `prompt ⟨annotation⟩ response`, masking annotation tokens from the loss to preserve the pretrained distribution. In 0.3% of examples, we mask only tag values to stabilize formatting (Appendix C.3). Learning rates are tuned via validation perplexity (Appendix C.2). As an ablation, we run annotation-anchored post-training from Standard checkpoints to isolate the contribution of annotated pretraining (Appendix C.4).

**Inference.** We generate outputs across sampling temperatures from 0.6 to 1.1. Unless otherwise stated, we report results at temperature 1, which provides a good balance between quality and diversity; the qualitative diversity–quality ordering is stable across temperatures on Stories (Figure 6). Note that for annotation-anchored models, annotation sampling is implicit: decoding from the model normally results in a sampled annotation. We retain only the response text for evaluation. Apart from temperature, the same decoding configuration is used for standard and annotation-anchored models, so any diversity gap is attributable to the trained model rather than to the sampler. All other inference hyperparameters are held fixed and reported in Appendix D.1.

### 5.2. Main results

**Annotation-anchoring mitigates collapse and restores positive scaling.** Figure 5 compares the semantic diversity of the generations of the base models (Stories), the SFT-trained models, and the annotation-anchored models across model scale. Mirroring the conclusion from Section 3, the base models become more semantically diverse as they scale. In contrast, standard SFT reduces diversity, and this reduction worsens with scale. However, annotation-anchored models reverse the trend, giving us our main result.

> **Main result**
>
> **Annotation-anchoring largely prevents post-training diversity collapse.** Across all four benchmarks, it keeps post-trained diversity much closer to the base model than standard SFT (STORIES), restores *positive* scaling of diversity with model size (NOVELTYBENCH and WILDCHAT), and broadly improves diversity.

For STORIES, the 2.5B-parameter annotation-anchored model closes the semantic diversity gap with the base model by roughly 85%, yielding a 6× reduction in diversity collapse compared to standard SFT. These results establish that diversity collapse is not an unavoidable consequence of post-training, but a distributional artifact that can be mitigated by anchoring the semantics.

**Annotation-anchoring preserves task performance.** The diversity gains of annotation-anchored training do not come at the cost of standard task performance. Table 3 reports zero-shot accuracy on a suite of reasoning and knowledge benchmarks (ARC, BoolQ, COPA, HellaSwag, OpenBookQA, PIQA, SciQ, WinoGrande); annotation-anchored and standard SFT models track each other closely at every scale, with average accuracies of 54.5% vs. 53.1% at 0.6B, 56.6% vs. 57.7% at 1B, and 62.3% vs. 62.4% at 2.5B—differences of at most about a point. The same pattern holds

| Size | Model Type | ARC Challenge | ARC Easy | BoolQ | COPA | Hella-Swag | Open-BookQA | PIQA | SciQ | Wino-Grande | Average |
|------|-----------|------|------|------|------|------|------|------|------|------|------|
| 0.6B | Annotated | 28.9 | 53.1 | 61.4 | 73.0 | 40.1 | 30.8 | 65.9 | 85.0 | 52.3 | 54.5 |
|      | SFT       | 28.4 | 53.6 | 55.4 | 68.0 | 41.1 | 31.4 | 65.7 | 81.5 | 52.3 | 53.1 |
| 1B   | Annotated | 31.5 | 57.1 | 62.4 | 68.0 | 45.3 | 33.8 | 68.2 | 87.8 | 55.0 | 56.6 |
|      | SFT       | 30.2 | 59.6 | 61.7 | 76.0 | 46.5 | 33.4 | 69.1 | 88.7 | 54.5 | 57.7 |
| 2.5B | Annotated | 36.1 | 65.8 | 65.5 | 78.0 | 55.0 | 36.4 | 70.8 | 92.7 | 59.9 | 62.3 |
|      | SFT       | 37.7 | 67.1 | 57.2 | 81.0 | 55.8 | 37.0 | 72.3 | 92.6 | 61.0 | 62.4 |

*Table 3.* Zero-shot accuracy on a suite of reasoning and knowledge benchmarks (ARC, BoolQ, COPA, HellaSwag, OpenBookQA, PIQA, SciQ, WinoGrande) for standard SFT and annotation-anchored models across model sizes. Average accuracy of the two pipelines tracks within roughly a point at every scale, indicating that annotation anchoring preserves benchmark task performance.

| | 0.6B | 1B | 2.5B |
|---|---|---|---|
| Standard | 11.3% | 19.4% | 35.4% |
| Annotated | 10.8% | 18.9% | 36.4% |

*Table 4.* Accuracy on grade-school mathematics (GSM8k) for standard SFT and annotation-anchored models across model sizes. The 2.5B annotation-anchored model attains 36.4% versus 35.4% for standard SFT, with the smaller scales similarly close, indicating that anchoring preserves performance on structured-reasoning tasks.

| Setting | Model size | | |
|---------|------|------|------|
|         | 0.6B | 1B | 2.5B |
| Base model (Standard) | 8.24 | 8.47 | 8.56 |
| Base model (Annotated) | 8.51 | 8.27 | 8.27 |
| Standard SFT | 6.86 | 6.64 | 6.88 |
| Annotation-anchored | 7.87 | 7.67 | 8.30 |
| + w/ standard pretraining | 7.21 | 7.06 | 7.07 |
| + w/o inference-time annotations | 6.26 | 6.48 | 7.01 |

*Table 5.* Ablations for semantic anchoring across scale: base models under standard vs annotated pretraining, standard SFT, and the full annotation-anchored pipeline, together with two ablations of that pipeline that drop annotated pretraining or that skip annotation sampling at inference. Removing either component sharply reduces diversity at every scale, showing that both annotated pretraining and inference-time annotation sampling are necessary for the full effect.

on structured-reasoning tasks: on grade-school mathematics (Table 4), the 2.5B annotation-anchored model attains 36.4% on GSM8k versus 35.4% for standard SFT, again within roughly a point and consistent across the smaller model sizes. This indicates that anchoring the semantic marginal preserves the conditional response behavior that drives benchmark accuracy.

**Annotation-anchoring improves the diversity–quality frontier.** We next evaluate the diversity–quality tradeoff. Figure 6 plots diversity versus judged quality across sampling temperatures on Stories. Standard SFT exhibits a steep tradeoff: increasing temperature degrades judged quality, and beyond a certain point, does not improve the diversity of the generations. Annotation-anchored models improve the Pareto frontier, achieving much higher diversity at a given quality level. We report corresponding curves for dialog benchmarks in Appendix G.1, and provide representative generation examples in Table 2.

## 5.3. Ablations and considerations

We have so far shown annotation-anchored post-training to be an effective strategy to train models that can natively produce diverse generations. However, the strategy presents additional complexity and computational requirements beyond the traditional pretraining and post-training setup. We ablate two of the key changes: the requirement to *pretrain with annotations*, and the requirement to *generate annotations at inference time*.

**Annotated pretraining is necessary for semantic anchoring.** To isolate whether semantic anchoring arises from annotated pretraining itself, we evaluate models trained with annotation-anchored post-training but initialized from Standard pretrained checkpoints. Table 5 compares standard SFT, full annotation-anchored training, and annotation-anchored post-training without annotated pretraining. Without annotated pretraining, diversity remains close to standard SFT across model sizes and well below the full annotation-anchored pipeline. This shows that semantic anchoring is established during pretraining and cannot be recovered by post-training alone, even with annotation supervision.

**Sampling annotations at inference is necessary for diverse generations.** We test whether sampling annotations during inference is required to realize diversity gains. We compare annotation-anchored models that generate responses with and without sampling annotations. Generating without first sampling annotations yields substantially lower diversity, despite identical training (Table 5). Conditioning on sampled semantic variables at inference is therefore essential for expressing the diversity that anchoring preserves

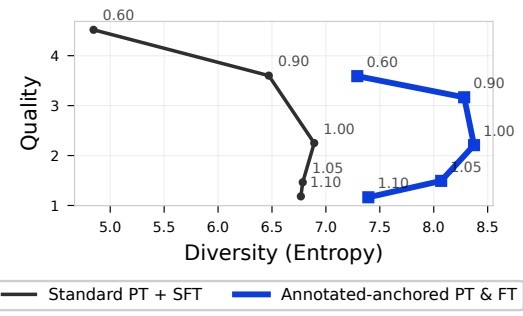

*Figure 6.* Diversity–quality tradeoff across sampling temperatures (point labels indicate the sampling temperature) on STORIES. Annotation-anchored generation improves the Pareto frontier, achieving higher diversity at comparable judged quality. We report corresponding curves for dialog benchmarks in Appendix G.1.

during training.

### 5.4. Controlled study: assessing the role of post-training data

*Annotation-anchored training* explicitly aims to optimize the model so that the generation distribution has the anchor distribution over the semantic variable given by the pretraining distribution $R(z \mid x)$, while updating the posterior distribution to match that of the post-training distribution $Q^\star(y \mid x, z)$. Perfectly optimizing this objective would yield generations with semantic entropy that is invariant to the semantic entropy of the post-training distribution. By contrast, standard SFT will collapse the semantic distribution to that of the post-training distribution, $R^\star(z \mid x)$. In this section, we test directly whether anchoring decouples generation diversity from the semantic entropy of the post-training data.

We use *SimpleStories*, a synthetic dataset of short stories labeled with semantic attributes including *topic* and *persona*; the full dataset contains 2M examples (Finke et al., 2025); see Appendix F for details. We construct fixed-size training subsets (∼200K examples) with varying semantic entropy by restricting attribute values. For topics, we select subsets where the topic is uniform over either 5, 14, or 48 values, yielding dataset entropies of $\log 5$, $\log 14$, and $\log 48$, respectively. For personas, we use 8, 12, and 23. We train standard SFT and annotation-anchored *2.5B*-parameter models on these subsets and sample generations at temperature 1. An LLM judge (Qwen3-30B-A3B-Instruct) assigns topic and persona labels to generated stories, enabling measurement of output entropy.

**Annotation-anchoring reduces sensitivity to dataset entropy.** Figure 7 shows that standard SFT tracks the dataset entropy: narrowing the training distribution narrows generation diversity. Annotation-anchored training is substantially less sensitive, preserving higher output diversity even when

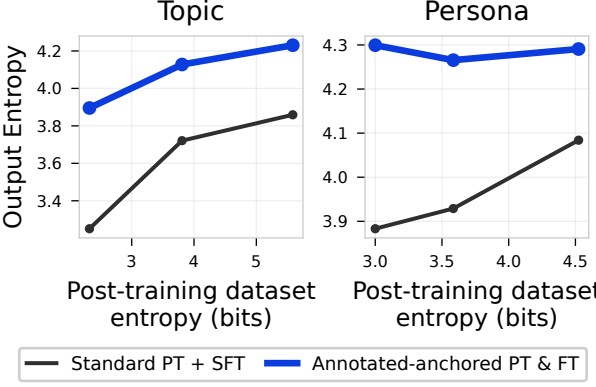

*Figure 7.* Controlled study on SIMPLESTORIES using our *2.5B*-parameter models: output semantic entropy as a function of post-training dataset entropy under standard SFT versus annotation-anchored training. *Annotation-anchored training* has substantially lower sensitivity to the post-training dataset entropy than SFT; for the PERSONA category, annotation-anchoring is essentially invariant to the entropy of the training dataset.

post-training data is narrow, and is essentially invariant to dataset entropy for personas. This shows how anchoring disentangles post-training semantics from generation diversity by preserving the pretraining prior.

## 6. Conclusion

Post-training improves instruction-following, but we show it can also induce *semantic mode collapse*: post-trained models become semantically homogeneous, and this collapse can *worsen with scale*. To mitigate this, we proposed *annotation-anchored training*, which factorizes post-training into what should *change* (the semantics-conditional response behavior) and what should *be preserved* from pretraining (a high-entropy semantic marginal). Empirically, anchoring largely prevents diversity collapse, restores positive scaling of semantic diversity, and improves the diversity–quality frontier, while a controlled study shows it reduces sensitivity to the semantic entropy of post-training data.

More broadly, our framework addresses a modern angle on continual-learning that is relevant for large-language models: the goal of post-training is not only to prevent performance degradation while adapting to new tasks, but also to retain desirable *distributional properties* from pretraining. This viewpoint suggests a broader impact: whenever post-training risks overwriting a distributional quantity that we wish to preserve, annotation-anchored training provides a simple template for updating conditional behavior while keeping the desired marginal pinned to its original pretraining distribution.

## Impact Statement

This paper presents work whose goal is to advance the field of Machine Learning. There are many potential societal consequences of our work, none of which we feel must be specifically highlighted here.

## Acknowledgements

This material is based upon work supported by the National Science Foundation Graduate Research Fellowship under Grant No. DGE2140739. Any opinions, findings, conclusions, or recommendations expressed in this material are those of the authors and do not necessarily reflect the views of the National Science Foundation.

We gratefully acknowledge support from Apple, Google, Jane Street, and the National Science Foundation.

The authors would like to thank Josh Susskind, Vimal Thilak, David Berthelot, Shuangfei Zhai, Ziqian Zhong, Gaurav Ghosal, Lawrence Feng, and Suhas Kotha for helpful discussions and feedback throughout the project.

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

# A. Pretraining Details

## A.1. Model Architecture

We train models with 0.6B, 1B, and 2.5B parameters using the OLMo-2 architecture (OLMo et al., 2024). The tokenizer used across all experiments is `allenai/OLMo-2-0425-1B-Instruct`, which provides consistent tokenization across model scales.

## A.2. Pretraining Data

Our pretraining corpus is derived from the Dolma (Soldaini et al., 2024) and Dolmino data mixes, but with a custom mixture tailored to our experiments. The data mixture consists of nine primary sources. Table 6 shows the composition of our pretraining data.

*Table 6.* Pretraining data mixture. Token counts are in billions (B). Proportions are relative to total tokens in the mixture.

| Source | Total Tokens (B) | Proportion |
|---|---|---|
| DCLM | 24.12 | 40.2% |
| Reddit | 7.16 | 11.9% |
| Tulu_Flan | 5.75 | 9.6% |
| CC-News | 5.31 | 8.8% |
| Books | 4.84 | 8.1% |
| Wikiref | 4.44 | 7.4% |
| Wikipedia | 3.62 | 6.0% |
| OpenWebMath | 2.52 | 4.2% |
| Pes2o | 2.08 | 3.5% |
| **Total** | 59.85 | 100% |

We train each model for a Chinchilla-optimal (Hoffmann et al., 2022) number of tokens with a scaling factor of 20:

- **0.6B model**: 12 billion tokens

- **1B model**: 20 billion tokens

- **2.5B model**: 50 billion tokens

## A.3. Pretraining Hyperparameters

Table 7 summarizes the pretraining hyperparameters. Learning rates are tuned per model size following established scaling laws.

*Table 7.* Pretraining hyperparameters. All models share common settings except for learning rate and total training tokens.

| Hyperparameter | Value |
|---|---|
| Sequence length | 2,048 |
| Global batch size (tokens) | 524,288 |
| Optimizer | AdamW |
| Adam $\beta_1, \beta_2$ | 0.9, 0.98 |
| Weight decay | 0.1 |
| Max gradient norm | 1.0 |
| Learning rate schedule | Cosine decay |
| Warmup steps | 2,000 |
| *Per-model learning rates:* | |
| 0.6B | $4 \times 10^{-3}$ |
| 1B | $3 \times 10^{-3}$ |
| 2.5B | $2 \times 10^{-3}$ |

### A.4. Compute Requirements

Pretraining the 2.5B parameter model requires approximately **2 days** on a single 8×H100 node (80GB per GPU). Smaller models train proportionally faster due to reduced token counts and model sizes.

## B. Annotation Details

### B.1. Annotator Model

We use **Qwen3-30B-A3B-Instruct** (Yang et al., 2025) as our annotation model for both pretraining and post-training data.

### B.2. Pretraining Annotation Procedure

For pretraining, we annotate documents at the sub-document level. Each document is split on double newlines ($\n\n$), and each resulting chunk is annotated independently. This procedure teaches the model a high-entropy conditional distribution over semantic annotations given context.

The annotation prompt template for pretraining is:

```
Pretraining Annotation Prompt

You are an expert annotator.  Read the following document and write a short set of
informative tags in the following format:  tag_type:tag_value tag_type:tag_value ...

The tag_type should be one or more of the following that is most relevant to the
document:  topic, domain, language, style, sentiment, action, entity, location, time,
etc.

Document:
"""{text}"""

Do not write any text other than the annotations.

Annotation:
```

### B.3. Post-Training Annotation Procedure

For post-training (TULU3 SFT data), we annotate each response using the final assistant message. The annotation prompt template is:

```
Post-Training Annotation Prompt

You are an expert annotator.  Read the following document and write a short set of
informative tags in the following format:  tag_type:tag_value tag_type:tag_value ...

The tag_type should be one or more of the following that is most relevant to the
document:  topic, domain, language, style, sentiment, action, entity, location, time,
etc.

Document:
"""{last_message}"""

Do not write any text other than the annotations.

Annotation:
```

### B.4. Annotation Schema

Annotations are serialized as variable-length tags of the form `<key>:<value>`, with a variable number of tags per document or response. Common tag types include:

- `topic`: The main subject matter (e.g., `topic:machine_learning`)
- `domain`: The broader field or category (e.g., `domain:science`)

- `entity`: Named entities mentioned (e.g., `entity:Einstein`)

- `action`: Key actions or events (e.g., `action:discovery`)

- `location`: Geographic references (e.g., `location:Paris`)

- `time`: Temporal references (e.g., `time:1920s`)

- `sentiment`: Emotional tone (e.g., `sentiment:positive`)

- `style`: Writing style (e.g., `style:formal`)

- `language`: Language of the text (e.g., `language:English`)

### B.5. Annotation Compute Requirements

Generating annotations for the full pretraining corpus requires approximately **3 days** on an $8\times$H100 node. This cost is *fixed* with respect to model scale—the same annotations can be reused for training models of any size. While annotation is computationally expensive, the amortized cost decreases as model scale increases, and we expect simpler annotation schemes could reduce this overhead (though we leave this exploration for future work).

## C. Post-Training Details

### C.1. Post-Training Dataset

We use the TULU3 SFT mixture (Lambert et al., 2024) for post-training, which contains 939,344 samples from diverse sources including:

- CoCoNot (10,983 prompts)

- FLAN v2 (89,982 prompts)

- No Robots (9,500 prompts)

- OpenAssistant Guanaco (7,132 prompts)

- NuminaMath-TIR (64,312 prompts)

- Tulu 3 WildGuardMix (50,000 prompts)

- Tulu 3 WildJailbreak (50,000 prompts)

- And additional synthetic and curated data

This dataset covers core skills including general instruction-following, knowledge recall, mathematics, coding, precise instruction-following, and safety.

### C.2. Post-Training Hyperparameters

Table 8 summarizes the post-training hyperparameters.

Learning rates are selected based on validation perplexity on a held-out portion of the TULU dataset. We train for a single epoch, as additional epochs are known to reduce generation diversity (Song et al., 2024).

### C.3. Annotation Loss Masking

During annotation-anchored post-training, we mask the loss for both the prompt tokens and the annotation tokens, so that gradients primarily update the response conditional $Q^\star(y \mid x, z)$ without modifying the pretrained annotation distribution $R(z \mid x)$.

**Partial unmasking for format stabilization**: In 0.3% of training examples, we mask only the `<value>` portion of each annotation tag while keeping the `<key>:` portion in the loss. This partial masking stabilizes annotation formatting at inference time by reinforcing the structural pattern of annotations without collapsing their semantic diversity.

*Table 8.* Post-training hyperparameters. Learning rate is selected via validation perplexity from the candidate set.

| Hyperparameter | Value |
|---|---|
| Batch size (sequences) | 512 |
| Training epochs | 1 |
| Max sequence length | 512 |
| Learning rate schedule | Cosine decay |
| Warmup ratio | 0.1 |
| Weight decay | 0.0 |

*Learning rate candidates (all model sizes):*
$$\{2 \times 10^{-4}, 4 \times 10^{-4}, 8 \times 10^{-4}\}$$

### C.4. Standard SFT Ablation

For the standard pretraining ablation (annotation-anchored post-training from standard pretrained checkpoints), we apply the same annotation-anchored post-training procedure but initialize from checkpoints that were pretrained *without* annotations. In this ablation, all annotation tokens remain unmasked during post-training. This isolates the contribution of annotated pretraining to semantic anchoring.

## D. Inference Details

### D.1. Sampling Configuration

Unless otherwise specified, all generations use the following sampling configuration:

*Table 9.* Default inference hyperparameters.

| Hyperparameter | Value |
|---|---|
| Temperature | 1.0 (varied 0.6–1.1 in ablations) |
| Top-$p$ (nucleus sampling) | 1.0 |
| Max new tokens | 512 |

For annotation-anchored models, annotation sampling is implicit: decoding from the model normally results in a sampled annotation, followed by a response conditioned on that annotation. We retain only the response text for evaluation.

### D.2. Sampling Procedures for Instruction-Tuned Models

We evaluate three sampling procedures for instruction-tuned models in Section 3:

1. **Direct**: Standard prompting where the model generates a single response to the prompt.

2. **Brainstorm**: We first prompt the model to generate "ideas" or a plan, then condition the final response on these generated ideas. This approach has been given multiple names, including Plan-and-Write, and Intent-Factored Generation (Yao et al., 2019; Ahmed et al., 2025).

3. **Multiple**$(n)$: We sample $n$ responses within the same context, including an explicit instruction to promote diversity across responses. We evaluate $n \in \{2, 8, 32\}$.

**Entropy aggregation.** For *direct* and *brainstorm* prompting, we compute semantic entropy $\widehat{H}$ (Appendix E.1) over the set of final sampled responses. For *multiple*$(n)$, each prompt invocation produces $n$ responses within one shared context; we treat all produced responses across all invocations as samples and compute $\widehat{H}$ over the pooled set. Equivalently, $\widehat{H}$ estimates the entropy of the label of a uniformly random response drawn from the multiset of all generated responses.

# E. Evaluation Details

## E.1. Semantic Diversity Metrics

### E.1.1. STORIES BENCHMARK: SEMANTIC ENTROPY

For the STORIES benchmark, we compute semantic entropy using LLM-based semantic labeling. Each generated story is mapped to semantic labels via an LLM judge (Qwen3-30B-A3B-Instruct), and we compute the entropy of the empirical distribution over labels.

We use the following semantic categories:

*Table 10.* Semantic categories for the STORIES benchmark.

| Category | Description |
|---|---|
| Summary | One-word summary of the story |
| Main Character | Name of the main character |
| Location | Primary setting (specific place) |
| Genre | Genre classification |
| Narrative Structure | Structural pattern of the narrative |
| Cultural Context | Cultural or mythic context |
| Worldbuilding Flavor | Worldbuilding characteristics |
| Imagery | Dominant imagery palette |

Semantic entropy is computed as the average entropy across all categories:

$$H_{\text{semantic}} = \frac{1}{|C|} \sum_{c \in C} H(p_c) \tag{4}$$

where $C$ is the set of semantic categories and $p_c$ is the empirical distribution of labels for category $c$.

### E.1.2. DIALOG BENCHMARKS: EMBEDDING DISSIMILARITY

For NOVELTYBENCH (Zhang et al., 2025b), WILDCHAT (Zhao et al., 2024), and INFINITYCHAT (Jiang et al., 2025), we measure diversity via mean pairwise cosine dissimilarity of response embeddings. Embeddings are computed using **Qwen3-Embedding-0.6B** (Yang et al., 2025), a text embedding model from the Qwen3 Embedding series with 0.6B parameters.

Embedding dissimilarity is computed as:

$$D = 1 - \frac{2}{n(n-1)} \sum_{i<j} \cos(e_i, e_j) \tag{5}$$

where $e_i$ are the embedding vectors of generated responses.

## E.2. LLM Judge Prompts

We use Qwen3-30B-A3B-Instruct as the LLM judge for both semantic labeling and quality evaluation. Below are the exact prompts used for each semantic category:

**Summary**

```
You are an expert in reading comprehension.  You summarize the story in *exactly
one word*.  Even if the summary would be better as a phrase, use exactly one word
anyways.  Be concise and be very specific to the story.  Do not output any other
text.
Story:  {output}
```

### Main Character Named

You are an expert in reading comprehension.  You identify the main character of the story in *exactly one word*.  If the main character has a multi-word name, output the main word only.  Be concise and be very specific to the story.  Do not output any other text.

Story: {output}

### Location

You are an expert in reading comprehension.  You identify the primary setting of the story (the specific place) in *exactly one word*.  If the location has multiple words, output the main word only.  Be concise and be very specific to the story.  Do not output any other text.

Story: {output}

### Genre

You are an expert in reading comprehension.  You describe the genre of the story in exactly one word.  If the genre has multiple words, output the main word only.  Be concise and be very specific to the story.  Do not output any other text.

Story: {output}

### Narrative Structure

You are an expert in reading comprehension.  You describe the narrative structure in exactly one word.  If the narrative structure has multiple words, output the main word only.  Be concise and be very specific to the story.  Do not output any other text.

Story: {output}

### Cultural Context

You are an expert in reading comprehension.  You describe the cultural or mythic context in exactly one word.  If the cultural context has multiple words, output the main word only.  Be concise and be very specific to the story.  Do not output any other text.

Story: {output}

### Worldbuilding Flavor

You are an expert in reading comprehension.  You describe the worldbuilding flavor in exactly one word.  If the worldbuilding flavor has multiple words, output the main word only.  Be concise and be very specific to the story.  Do not output any other text.

Story: {output}

### Imagery

You are an expert in reading comprehension.  You describe the dominant imagery palette in exactly one word.  If the imagery palette has multiple words, output the main word only.  Be concise and be very specific to the story.  Do not output any other text.

Story: {output}

### E.3. Quality Evaluation

Response quality is evaluated using the same LLM judge (Qwen3-30B-A3B-Instruct) with a Likert-scale rating prompt. We report mean quality scores across generated samples.

We score responses on a 1–5 scale across coherence, prompt relevance, and narrative engagement. While capability benchmarks (e.g., MMLU) would assess factual knowledge retention, our quality metric specifically targets generation characteristics relevant to creative and open-ended tasks where diversity matters most. We leave capability benchmark evaluation to future work, noting that *annotation-anchored training* modifies only how semantic plans are sampled, not the underlying knowledge representations.

## F. Controlled Study: SimpleStories Details

### F.1. Dataset Description

SimpleStories (Finke et al., 2025) is a synthetic dataset of 2 million short stories generated using GPT-4o-mini. Each story is annotated with semantic attributes including topic and persona, enabling controlled experiments on the relationship between post-training data entropy and generation diversity.

### F.2. Subset Construction

We construct fixed-size training subsets ($\sim$200K examples each) with varying semantic entropy by restricting the allowed values for topic and persona attributes.

#### F.2.1. TOPIC CATEGORIES

We use three topic subsets with increasing entropy:

**Topic-5**    (5 values, $\log_2 5 \approx 2.32$ bits):

    space exploration, gardens, talking animals, treasure hunts, hidden treasures

**Topic-14**    (14 values, $\log_2 14 \approx 3.81$ bits):

    space exploration, gardens, talking animals, treasure hunts, hidden treasures, the sky, fairy tales, the arts, secret societies, outer space, school life, riddles, undercover missions, seasonal changes

**Topic-48**    (48 values, $\log_2 48 \approx 5.58$ bits):

    space exploration, gardens, talking animals, treasure hunts, hidden treasures, the sky, fairy tales, the arts, secret societies, outer space, school life, riddles, undercover missions, seasonal changes, invisibility, holidays, mystical creatures, dream worlds, living objects, subterranean worlds, enchanted forests, dinosaurs, shape-shifting, bygone eras, underwater adventures, unusual vehicles, a deadline or time limit, superheroes, island adventures, robots and technology, mysterious maps, alien encounters, sibling rivalry, magical lands, royal kingdoms, virtual worlds, cultural traditions, lost civilizations, miniature worlds, sports, time travel, haunted places, magical objects, lost cities, fantasy worlds, pirates, giant creatures, snowy adventures

#### F.2.2. PERSONA CATEGORIES

We use three persona subsets with increasing entropy:

**Persona-8**    (8 values, $\log_2 8 = 3.0$ bits):

    an innocent author, someone who loves order and structure, a hopeless romantic, a hurt ill-intentioned person, a wise old person who wants to teach the young, a father, a powerful leader, the everyman

**Persona-12**    (12 values, $\log_2 12 \approx 3.58$ bits):

    an innocent author, someone who loves order and structure, a hopeless romantic, a hurt ill-intentioned person, a wise old person who wants to teach the young, a father, a powerful leader, the everyman, a philosopher, an explorer archetype, someone who wants to prove a point, a pedant

**Persona-23**  (23 values, $\log_2 23 \approx 4.52$ bits):

> an innocent author, someone who loves order and structure, a hopeless romantic, a hurt ill-intentioned person, a wise old person who wants to teach the young, a father, a powerful leader, the everyman, a philosopher, an explorer archetype, someone who wants to prove a point, a pedant, someone curious, a cruel person, an academic, a jester archetype, a poet, someone evil, a child, a mother, a moralistic teacher, a rebellious author, the oppressed

### F.3. Experimental Protocol

For each subset (topic and persona), we:

1. Sample $\sim$200K examples uniformly over the allowed attribute values

2. Train both standard SFT and annotation-anchored 2.5B-parameter models

3. Generate stories at temperature 1.0

4. Use the LLM judge (Qwen3-30B-A3B-Instruct) to assign topic and persona labels to generated stories

5. Compute output entropy over the assigned labels

## G. Additional Results

### G.1. Diversity–Quality Curves for Dialog Benchmarks

Figure 8 shows the diversity–quality tradeoff across sampling temperatures for the dialog benchmarks (NOVELTYBENCH, WILDCHAT, INFINITYCHAT). Consistent with the STORIES results in the main paper, annotation-anchored models improve the Pareto frontier across all benchmarks.

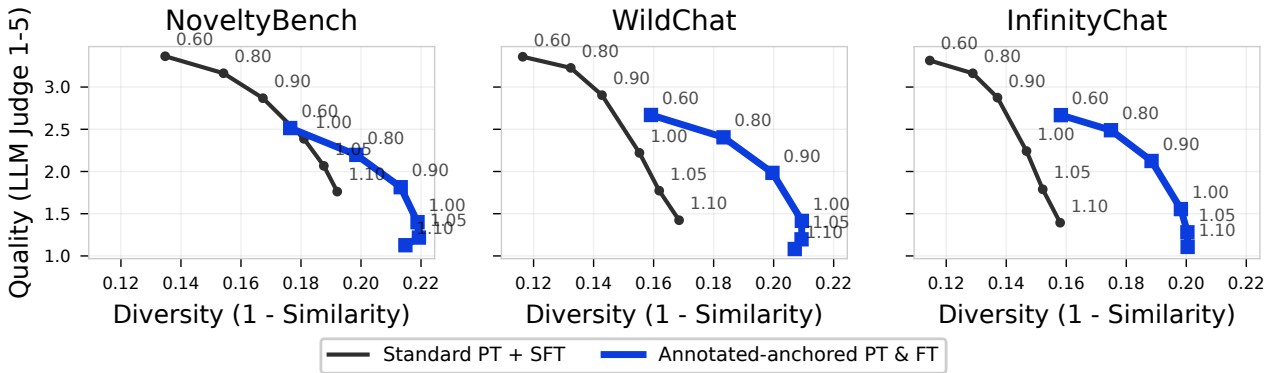

*Figure 8.* Diversity–quality tradeoff across sampling temperatures for dialog benchmarks.

### G.2. Diversity Judge Robustness

To confirm that our diversity findings are not an artifact of a single LLM judge, we re-evaluate the 2.5B Standard SFT and annotation-anchored models on STORIES using GPT-5.4 as an independent diversity judge across sampling temperatures. The relative ordering matches the Qwen3-30B-A3B-Instruct judge used in the main paper: the annotation-anchored model is judged substantially more diverse than the Standard model across the full temperature range (Table 11). This indicates that the diversity gap we observe is not an artifact of judge selection.

| Model | T=0.6 | T=0.9 | T=1.0 | T=1.05 | T=1.1 |
|---|---|---|---|---|---|
| Standard (2.5B) | 5.07 | 6.88 | 7.39 | 6.94 | 6.41 |
| Annotated (2.5B; ours) | 7.13 | 8.36 | 8.1 | 7.39 | 6.8 |

*Table 11.* Diversity scores from GPT-5.4 as judge on STORIES across temperatures (2.5B models). The Annotated model is judged substantially more diverse than the Standard model across the temperature range.

