# OpenReview forum: "Annotations Mitigate Post-Training Mode Collapse"
_ICML.cc/2026/Conference — ICML 2026 regular_

### Official Review · Reviewer_r4Ha · 2026-03-09

**Soundness:** 2
**Presentation:** 3
**Significance:** 3
**Originality:** 3
**Overall Recommendation:** 3
**Confidence:** 4

**Summary:**

This paper addresses the semantic mode collapse triggered by supervised fine-tuning (SFT), a phenomenon where model outputs become increasingly homogeneous as model scale increases. The authors propose annotation-anchored training, which decouples the high-entropy semantic distribution of pre-training from the conditional response behavior of post-training. By augmenting pre-training with structured semantic tags and masking these tags' loss during SFT, the model's semantic marginal is "anchored" to its diverse pre-trained state. At inference, the model samples a semantic plan before generating a response. This approach successfully reverses the inverse-scaling trend of diversity, reducing collapse by up to 6x while significantly improving the diversity-quality Pareto frontier across multiple open-domain benchmarks.

**Compliance With Llm Reviewing Policy:**

Affirmed.

**Final Justification:**

Due to concerns regarding deterministic tasks such as mathematical reasoning, and the corresponding potential for methodological optimization within these areas, I will maintain my current score.

**Key Questions For Authors:**

- Pre-training typically involves long-form text, whereas post-training is structured around specific tasks (input-output pairs). How does this format gap between chunked annotations in pre-training and prompt-response pairs in SFT affect the model’s ultimate performance?

- Is it possible to apply the annotation-anchored approach via continual pre-training on a smaller, high-quality subset of data instead of modifying the entire full pre-training stage?

- For structured data like mathematics or code, is it necessary to apply the same level of semantic annotation, or would it potentially interfere with the model's ability to learn logical sequences?

**Limitations:**

Yes

**Strengths And Weaknesses:**

Pros:
- The paper provides a thorough analysis of semantic mode collapse, which is a pivotal issue for the performance of LLMs in open-domain and creative generation tasks.
- The proposed annotation-anchored training is conceptually sound and provides a mechanistically clear way to decouple semantic variety from instruction-following behaviors.


Cons:
- My primary concern lies in the applicability of this method to non-creative tasks, such as mathematical reasoning or coding. It remains unclear whether this approach would lead to performance degradation in accuracy-driven benchmarks, especially considering the slight quality drops already observed in some dialogue tasks.
- The requirement to annotate the entire pre-training corpus is substantial. Given the massive scale of modern pre-training datasets, generating high-quality semantic tags for every document introduces significant computational cost and resource consumption.

---

> ### Author Rebuttal · Authors · 2026-03-31
>
> Dear Reviewer,
>
> Thank you for your thoughtful feedback. We respond inline below.
>
> > My primary concern lies in the applicability of this method to non-creative tasks, such as mathematical reasoning or coding. It remains unclear whether this approach would lead to performance degradation in accuracy-driven benchmarks, especially considering the slight quality drops already observed in some dialogue tasks.
>
> We added evaluation on a standard set of LLM benchmarks for small models (e.g., the OLMo-1B evaluation suite):
>
> |Size|Model Type|arc_challenge|arc_easy|boolq|copa|hellaswag|openbookqa|piqa|sciq|winogrande|Average|
> |-|-|-|-|-|-|-|-|-|-|-|-|
> |0.6B|Annotated|28.9%|53.1%|61.4%|73.0%|40.1%|30.8%|65.9%|85.0%|52.3%|54.5%|
> |0.6B|SFT|28.4%|53.6%|55.4%|68.0%|41.1%|31.4%|65.7%|81.5%|52.3%|53.1%|
> |-|-|-|-|-|-|-|-|-|-|-|-|
> |1B|Annotated|31.5%|57.1%|62.4%|68.0%|45.3%|33.8%|68.2%|87.8%|55.0%|56.6%|
> |1B|SFT|30.2%|59.6%|61.7%|76.0%|46.5%|33.4%|69.1%|88.7%|54.5%|57.7%|
> |-|-|-|-|-|-|-|-|-|-|-|-|
> |2.5B|Annotated|36.1%|65.8%|65.5%|78.0%|55.0%|36.4%|70.8%|92.7%|59.9%|62.3%|
> |2.5B|SFT|37.7%|67.1%|57.2%|81.0%|55.8%|37.0%|72.3%|92.6%|61.0%|62.4%|
>
> Performance is essentially unchanged: 54.5/53.1 at 0.6B, 56.6/57.7 at 1B, and 62.3/62.4 at 2.5B for ours/baseline.
>
> > The requirement to annotate the entire pre-training corpus is substantial. Given the massive scale of modern pre-training datasets, generating high-quality semantic tags for every document introduces significant computational cost and resource consumption.
>
> We added a simulation of annotation cost relative to training cost. Annotating the corpus for a 70B model is only 1.5% of total training cost; at 1T scale it is 0.1%.
>
> Using KV caching, annotation requires roughly one forward pass over the corpus plus generated annotation tokens:
>
> $\text{annotation cost} \approx 2 \times (\text{active params}) \times (\text{original pretraining tokens} + \text{generated annotation tokens}), \; \text{training} \approx 6 \times (\text{params}) \times (\text{tokens})$.
>
> Using Qwen3-30B-A3B as annotator, we obtain:
>
> | |8B|30B|70B|300B|1T|
> |-|-|-|-|-|-|
> |Annotation share|12.1%|3.5%|1.5%|0.4%|0.1%|
> |Training share|87.9%|96.5%|98.5%|99.6%|99.9%|
>
> Thus, annotation is a very small fraction of total compute, especially at frontier scales. Similar model-based preprocessing is already common in pre-training, e.g., for data filtering [1] or rephrasing [2].
>
> > Pre-training typically involves long-form text, whereas post-training is structured around specific tasks (input-output pairs). How does this format gap between chunked annotations in pre-training and prompt-response pairs in SFT affect the model’s ultimate performance?
>
> Empirically, the format gap does not prevent transfer of the learned annotation prior. The full pipeline improves diversity substantially on Stories and dialogue benchmarks, suggesting that chunk-level annotated pre-training learns a useful conditional annotation distribution that transfers to prompt-response post-training.
>
> > Is it possible to apply the annotation-anchored approach via continual pre-training on a smaller, high-quality subset of data instead of modifying the entire full pre-training stage?
>
> This is a great question and an interesting avenue for future work. We expect our method to also work well as a continual pre-training stage, especially when the continual pre-training data is sufficiently large and diverse to learn a rich annotation distribution.
>
> In summary, we have added capability-benchmark results, quantified the annotation-compute tradeoff, clarified why the learned annotation prior can transfer across the pre-training / post-training format gap, and added a more explicit answer about structured domains such as math and code.
>
> We very much appreciate your review. We would appreciate it if you would consider raising your score in light of the new experiments and clarifications we have added.
>
> Thank you very much,
>
> Authors
>
>
> [1] Zhang et al., GPT-4o as the Gold Standard: A Scalable and General Purpose Approach to Filter Language Model Pretraining Data (2024)
>
> [2] Maini et al., Rephrasing the Web: A Recipe for Compute and Data-Efficient Language Modeling (2024)

---

> > ### Author Rebuttal · Reviewer_r4Ha · 2026-04-03
> >
> > Thank you for the detailed rebuttal and the additional experiments.
> >
> > While several of my earlier concerns have been addressed, I still have a question regarding the applicability of the annotation schema to structured/closed-domain tasks. For open-ended generation (stories, dialogue), sampling diverse entities or topics as anchors is natural and well-motivated. But for closed-domain tasks such as mathematical reasoning or code generation, the semantics of a correct response are heavily constrained by the input. Specifically:
> > - What would annotations look like in these domains — would an entire math problem simply receive a coarse annotation (e.g., `domain:math`), or would there be finer-grained semantic distinctions (e.g., distinguishing proof strategies, solution approaches, or sub-problem decompositions)?
> > - During inference on a math/code prompt, is there a risk that the model samples annotations that are irrelevant or even detrimental to the logical coherence of the response?
> >
> > I raise this because the pre-training corpus (Table 4) includes OpenWebMath (4.2%) and code-adjacent data, and the post-training mixture (TULU3) includes NuminaMath and coding tasks, yet the paper does not discuss how annotations interact with these structured domains. A brief analysis or discussion of annotation behavior on such data would strengthen the paper.

---

> > > ### Author Response · Authors · 2026-04-06
> > >
> > > This is a great question, and thanks for clarifying. We have added a new experiment to address your final concern: we evaluate our models on GSM8k and find that annotated training performs essentially equivalently to the standard SFT models. We find that annotations do not negatively impact the logical coherence of the responses.
> > >
> > > We tune over temperature (0.6–1.1) and report GSM8k accuracy. Here is a table:
> > >
> > > | | 0.6B | 1B | 2.5B |
> > > |---|---|---|---|
> > > | Vanilla | 11.3% | 19.4% | 35.4% |
> > > | Annotated | 10.8% | 18.9% | 36.4% |
> > >
> > > Since our focus was not on math, we did not prompt the pre-training annotations to include these math-specific “finer-grained semantic distinctions” that you mention (e.g., different approaches, subproblem decompositions, etc). However, these could easily be added to our pipeline and would be a promising avenue of future work to boost math performance!
> > >
> > > Nonetheless, to answer your question regarding what these annotations look like in these domains: we provide an example generation from the 2.5B annotated model for GSM8k as an example:
> > >
> > > Prompt:
> > > ```
> > > Janet’s ducks lay 16 eggs per day. She eats three for breakfast every morning and bakes muffins for her friends every day with four. She sells the remainder at the farmers' market daily for $2 per fresh duck egg. How much in dollars does she make every day at the farmers' market?
> > > ```
> > >
> > > Model response:
> > > ```
> > > <|annotate|>
> > > topic:duck products
> > > domain:agriculture
> > > time:every day
> > > entity:eggs
> > > action:producing
> > > entity:ducks
> > > <|annotate|>
> > > Janet's ducks lay 16 eggs per day.
> > > She eats 3 eggs for breakfast and bakes muffins with 4 eggs.
> > > So she has 16 - 3 - 4 = 13 - 4 = 9 eggs left.
> > > She sells the remainder at the farmers' market daily for $2 per fresh duck egg.
> > > So she makes 9 * $2 = $18 per day.
> > > Thus Janet makes \boxed{18} dollars every day at the farmers' market.
> > > ```
> > >
> > > We will add discussion of this, as well as our GSM8k results in the final revision.
> > >
> > > We would appreciate it if you would consider raising your score if this has addressed your concerns!

---

### Official Review · Reviewer_PTWZ · 2026-03-12

**Soundness:** 3
**Presentation:** 3
**Significance:** 3
**Originality:** 3
**Overall Recommendation:** 5
**Confidence:** 3

**Summary:**

This paper studies semantic mode collapse induced by post-training: while supervised fine-tuning improves instruction-following, it can also make generations semantically narrower, and the paper argues that this effect worsens with model scale. The core proposal is annotation-anchored training, which makes a semantic latent variable explicit via text annotations. During pre-training, the model is trained on annotation-document sequences so it learns a rich annotation distribution; during post-training, responses are conditioned on annotations but the annotation tokens are masked from the loss so the pre-trained semantic distribution is preserved; at inference, the model first samples annotations and then generates conditioned on them. Empirically, the paper reports that this substantially mitigates diversity collapse relative to standard SFT, restores positive scaling of diversity with model size, improves the diversity-quality frontier, and reduces sensitivity to the entropy of the post-training dataset in a controlled synthetic study.

**Compliance With Llm Reviewing Policy:**

Affirmed.

**Final Justification:**

My main concerns have been addressed during the rebuttal. The limitation that the method is only trained up to 2.5B parameters remains, but I view this as a reasonable compute/scope limitation rather than a flaw that undermines the main contribution, especially given the broader evidence that the underlying collapse phenomenon persists at larger scales. I will raise my score.

**Key Questions For Authors:**

Please refer to the weaknesses mentioned above. I would appreciate it if authors could elaborate on them and/or provide experimental results.

**Limitations:**

Yes.

**Strengths And Weaknesses:**

## Strengths

1 - The factorization of the generation distribution into a semantic marginal R(z|x) and a conditional Q*(y|x,z) is elegant and well-motivated. The insight that standard SFT collapses both when you only want to update the conditional is a genuinely useful conceptual contribution.

2 - The paper has a strong empirical story on the diagnostic side. Section 3 is convincing. Reproducing the inverse-scaling phenomenon from NoveltyBench across two model families (Qwen2.5, Llama 3), multiple prompting strategies, and showing the correlation between post-training loss and diversity loss (Figure 3) builds a solid empirical case.

3 - The paper is well-written with a clear narration.


---

## Weaknesses

1 - My main concern is that the entire motivation hinges on mode collapse worsening with scale, yet the proposed method is only evaluated on models up to 2.5B parameters pre-trained from scratch. The paper acknowledges this limitation (Section 6.1), but it's a significant gap because: (a) the method requires modifying pre-training, making it expensive to validate at realistic scales; (b) the positive scaling trend in Figure 4 is based on only three points (0.6B, 1B, 2.5B), which is insufficient to confidently extrapolate.


2 - Another concern is that there is no evaluation of downstream task performance/capabilities. The paper evaluates diversity and a Likert-scale quality metric from an LLM judge, but there is no evaluation on standard capability benchmarks (MMLU, GSM8K, HumanEval, etc.). The authors explicitly punt on this (Appendix E.3). This is a significant omission: if annotation-anchored training preserves diversity but degrades factual accuracy, coding ability, or reasoning, the method has a hidden cost. Reviewers need to know whether the annotation tokens "steal" capacity from content tokens during pre-training (since they replace content tokens to keep total FLOPs matched).

3 - Generating annotations with Qwen3-30B-A3B for the entire pre-training corpus takes ~3 days on 8×H100s. For the model scales studied here (≤2.5B, ≤50B tokens), this is already nontrivial. For frontier-scale corpora (10T+ tokens), this cost would be enormous. The paper hand-waves this as "amortized" but doesn't quantify the actual ratio. More importantly, the annotation quality is not validated.

4 - The primary Stories metric relies on an LLM judge assigning single-word labels across 8 categories, then computing entropy over those labels. This is a coarse discretization; the entropy measurement is upper-bounded by the granularity of the label space, and the quality of the labels depends entirely on the judge.


5 - The paper compares only against vanilla SFT. Natural baselines are missing: (a) KL-constrained SFT (e.g., with a KL penalty to the base model), which is the standard way to prevent distribution collapse; (b) entropy-regularized SFT (Zhang et al., 2024), which is cited but not compared against.

---

> ### Author Rebuttal · Authors · 2026-03-31
>
> Dear Reviewer,
>
> Thank you for your thoughtful feedback. We respond inline below.
>
> > My main concern is that the entire motivation hinges on mode collapse worsening with scale [...] (a) the method requires modifying pre-training [...]; (b) the positive scaling trend in Figure 4 is based on only three points (0.6B, 1B, 2.5B).
>
> We agree these are limitations, but believe the paper remains relevant.
>
> Regarding (b), compute constraints prevent training beyond 2.5B parameters. However, we evaluate Qwen and Llama models up to 70B and find that larger post-trained models consistently exhibit less diversity. While we cannot guarantee the positive trend for our method persists indefinitely, we believe demonstrating it at these scales is meaningful.
>
> > There is no evaluation of downstream task performance.
>
> We added evaluation on a standard suite of LLM benchmarks for small models:
>
> |Size|Model Type|arc_challenge|arc_easy|boolq|copa|hellaswag|openbookqa|piqa|sciq|winogrande|Average|
> |-|-|-|-|-|-|-|-|-|-|-|-|
> |0.6B|Annotated|28.9%|53.1%|61.4%|73.0%|40.1%|30.8%|65.9%|85.0%|52.3%|54.5%|
> |0.6B|SFT|28.4%|53.6%|55.4%|68.0%|41.1%|31.4%|65.7%|81.5%|52.3%|53.1%|
> |-|-|-|-|-|-|-|-|-|-|-|-|
> |1B|Annotated|31.5%|57.1%|62.4%|68.0%|45.3%|33.8%|68.2%|87.8%|55.0%|56.6%|
> |1B|SFT|30.2%|59.6%|61.7%|76.0%|46.5%|33.4%|69.1%|88.7%|54.5%|57.7%|
> |-|-|-|-|-|-|-|-|-|-|-|-|
> |2.5B|Annotated|36.1%|65.8%|65.5%|78.0%|55.0%|36.4%|70.8%|92.7%|59.9%|62.3%|
> |2.5B|SFT|37.7%|67.1%|57.2%|81.0%|55.8%|37.0%|72.3%|92.6%|61.0%|62.4%|
>
> Performance is essentially unchanged: 54.5/53.1 at 0.6B, 56.6/57.7 at 1B, and 62.3/62.4 at 2.5B for ours/baseline.
>
> > The paper hand-waves [annotation compute cost] as "amortized" but doesn't quantify the actual ratio.
>
> We added a simulation of annotation cost relative to training cost. Annotating the corpus for a 70B model is only 1.5% of total training cost; at 1T scale it is 0.1%.
>
> Using KV caching, annotation requires roughly one forward pass over the corpus plus generated annotation tokens:
>
> $\text{annotation cost} \approx 2 \times (\text{active params}) \times (\text{original pretraining tokens} + \text{generated annotation tokens}), ; \text{training} \approx 6 \times (\text{params}) \times (\text{tokens})$
>
> Using Qwen3-30B-A3B as annotator, we obtain:
>
> | |8B|30B|70B|300B|1T|
> |-|-|-|-|-|-|
> |Annotation share|12.1%|3.5%|1.5%|0.4%|0.1%|
> |Training share|87.9%|96.5%|98.5%|99.6%|99.9%|
>
> Thus, annotation is a very small fraction of total compute, especially at frontier scales. Related model-based corpus preprocessing is already common, e.g., for data filtering [1] or rephrasing [2].
>
> > The primary Stories metric relies on an LLM judge assigning single-word labels [...] This is a coarse discretization [...]
>
> We added an experiment using GPT-5.4 as annotator/evaluator. On Stories, we again observe a similar gain over standard SFT:
>
> |model_type|temp=0.6|temp=0.9|temp=1.0|temp=1.05|temp=1.1|
> |-|-|-|-|-|-|
> |Vanilla (2.5B)|5.07|6.88|7.39|6.94|6.41|
> |Annotated (2.5B; ours)|*7.13*|*8.36*|*8.1*|*7.39*|*6.8*|
>
> We will switch the paper’s results to a GPT-5.4-based annotator. For NoveltyBench and WildChat, we also evaluate mean embedding cosine dissimilarity and observe analogous improvements.
>
> > The paper compares only against vanilla SFT. Natural baselines are missing: (a) KL-constrained SFT [...]; (b) entropy-regularized SFT [...]
>
> We added a KL-regularized SFT baseline toward the base model, with regularization strength $\lambda$. On Stories, we evaluate both diversity and quality:
>
> | |Annotated|KL (λ=0.01)|KL (λ=0.1)|KL (λ=1.0)|
> |-|-|-|-|-|
> |Diversity|8.282|6.643|6.969|8.035|
> |Quality|3.167|3.38|3.193|2.394|
>
> With small-to-medium regularization, diversity remains substantially below annotation-anchored training. With strong regularization, diversity is still lower and quality degrades substantially. Thus, annotation-anchored training achieves a better diversity-quality tradeoff than KL regularization.
>
> ---
>
> In summary, we have added new results on capability retention, quantified the annotation-compute overhead, strengthened the discussion of evaluator dependence, and added a KL-constrained baseline. We will also explicitly acknowledge the remaining missing baselines and the lack of direct annotation-quality validation in the limitations.
>
> We very much appreciate your review. We would appreciate it if you would consider raising your score in light of the new experiments and clarifications we have added.
>
> Thank you very much,
>
> Authors
>
>
> [1] Zhang et al., GPT-4o as the Gold Standard: A Scalable and General Purpose Approach to Filter Language Model Pretraining Data (2024)
>
> [2] Maini et al., Rephrasing the Web: A Recipe for Compute and Data-Efficient Language Modeling (2024)

---

> > ### Author Rebuttal · Reviewer_PTWZ · 2026-04-03
> >
> > Thank you for the detailed rebuttal. My main concerns have been adequately addressed.
> > The limitation that the method is only trained up to 2.5B parameters remains, but I view this as a reasonable compute/scope limitation rather than a flaw that undermines the main contribution, especially given the broader evidence that the underlying collapse phenomenon persists at larger scales. I will raise my score.

---

### Official Review · Reviewer_XbLF · 2026-03-12

**Soundness:** 3
**Presentation:** 3
**Significance:** 3
**Originality:** 3
**Overall Recommendation:** 4
**Confidence:** 3

**Summary:**

This paper addresses semantic mode collapse in LLMs during the post-training phase. The authors note that supervised fine-tuning restricts generation diversity by confining the model to the narrow semantic distribution of the tuning data, a problem that actually worsens as model size increases. They introduce Annotation Anchored Training to separate response formatting from content semantics. The method modifies the pre-training phase by augmenting documents with automatically generated semantic tags, helping the model learn a broad semantic prior. During fine-tuning, the model learns to generate responses based on these tags, but the loss for the prompt and tags is masked. This masking prevents the tuning process from overwriting the diverse semantics learned earlier. At inference, the model first generates a semantic plan and then produces the final text.

Experimentally, the authors train custom models of various sizes up to 2.5B parameters on a mixed corpus and fine-tune them on a standard instruction dataset. They evaluate the models on open-ended story generation and multiple dialog benchmarks using metrics like LLM judged semantic entropy and embedding dissimilarity. The results show that the proposed approach maintains higher diversity levels than standard fine-tuning across all tested scales. It also restores a positive scaling trend where larger models produce more varied outputs while maintaining generation quality. A controlled study with synthetic data further demonstrates that the anchored models are less sensitive to the restricted semantic variety of the fine-tuning data.

**Compliance With Llm Reviewing Policy:**

Affirmed.

**Final Justification:**

The additional experiments provided in the rebuttal successfully addressed my major concerns #2 and #3 (but #1 remained), so I have raised my final score to 4.

**Key Questions For Authors:**

The premise relies on SFT data having a low-entropy semantic prior. If the community develops extremely high-quality and diverse post-training datasets in the future, do you believe this "inverse scaling" phenomenon will naturally disappear, or is it an inherent flaw of the standard SFT objective itself?

**Limitations:**

yes

**Strengths And Weaknesses:**

**Strengths**
1. The paper highlights a highly relevant and counterintuitive "inverse scaling" phenomenon where larger models suffer more from semantic mode collapse after post-training
2. The proposed Annotation-Anchored Training is intuitive, well-motivated, and demonstrates clear empirical effectiveness
3. The experimental design is rigorous, covering both pre-training and post-training phases, with benchmarks showing distinct trends that strongly support the claims

**Weaknesses**

Major concerns
1. As the authors discuss, the proposed method must be integrated during the pre-training phase to be fully effective; applying it only during post-training yields limited benefits. This makes the approach extremely resource-intensive and prevents practitioners from simply applying it to existing, powerful open-weight base models, significantly limiting the method's practical utility and broader impact.
2. The reliance on `Qwen3-30B-A3B-Instruct` as both the annotator and the LLM judge creates a potential methodological flaw. First, using a post-trained model to annotate pre-training data acts as a "low-pass filter," potentially forcing the high-entropy pre-training distribution into the model's familiar, lower-entropy semantic buckets, which contradicts the goal of capturing the full breadth of pre-training data. Second, and more importantly, training the target models (0.6B-2.5B) to learn from these specific tags and then using the exact same Qwen model to judge their semantic entropy and quality introduces potential preference bias and leakage. The models are essentially being distilled to mimic the judge's specific semantic manifold.
3. While the authors justify omitting standard factual or reasoning benchmarks (e.g., MMLU, GSM8K) by stating their method primarily affects generative planning, this omission is a notable issue. Without evaluating on these mainstream capability benchmarks, it is impossible to assess whether maintaining this semantic diversity incurs a significant "alignment tax" or trade-off regarding the model's core abilities.

Minor concern:

The requirement to auto-regressively generate semantic tags before outputting the actual response introduces an inevitable and minor increase in inference cost and latency

---

> ### Author Rebuttal · Authors · 2026-03-31
>
> Dear Reviewer,
>
> Thank you for your thoughtful feedback. We respond inline below.
>
> > The proposed method must be integrated during pre-training [...], making it resource-intensive and hard to apply to existing open-weight models.
>
> This is a fair concern. However, our goal is to understand how to intervene during pre-training to produce models that can later be post-trained without semantic mode collapse. We believe this is broadly relevant to the ML community especially those interested in understanding how best to pretrain models, since recent language models still exhibit substantial semantic collapse after post-training (Figure 2), so this remains an important open problem.
>
> > The reliance on Qwen3-30B-A3B-Instruct as both the annotator and the LLM judge creates a potential methodological flaw. [...] Training the target models (0.6B-2.5B) to learn from these specific tags and then using the exact same Qwen model to judge their semantic entropy and quality introduces potential preference bias and leakage.
>
> We added an experiment using GPT-5.4 as the annotator/evaluator to address this concern. On Stories (analogous to Figure 5), we observe a similar gain over standard SFT:
>
> | model_type       |   temp=0.6 |   temp=0.9 |   temp=1.0 |   temp=1.05 |   temp=1.1 |
> |:-----------------|-----------:|-----------:|-----------:|------------:|-----------:|
> | Vanilla (2.5B)          |       5.07 |       6.88 |       7.39 |        6.94 |       6.41 |
> | Annotated (2.5B; ours) |       *7.13* |       *8.36* |       *8.1*  |       *7.39* |       *6.8*  |
>
> Annotated-anchored training therefore continues to improve diversity under an independent evaluator. We will switch the paper’s results to a GPT-5.4-based annotator.
>
> > Using a post-trained model to annotate pre-training data may act as a "low-pass filter" [...]
>
> We added analysis of annotation diversity. Our annotations contain 152M unique values over 156M documents and 2.3B total key-value pairs, indicating that the annotation distribution itself is extremely broad.
>
> > Omitting standard factual/reasoning benchmarks [...] makes it unclear whether there is an "alignment tax."
>
> We added evaluation on a standard small-model benchmark suite:
>
> | Size | Model Type | arc_challenge | arc_easy | boolq | copa | hellaswag | openbookqa | piqa | sciq | winogrande | Average |
> |------|------------|------|------|------|------|------|------|------|------|------|------|
> | 0.6B | Annotated | 28.9% | 53.1% | 61.4% | 73.0% | 40.1% | 30.8% | 65.9% | 85.0% | 52.3% | 54.5% |
> | 0.6B | SFT | 28.4% | 53.6% | 55.4% | 68.0% | 41.1% | 31.4% | 65.7% | 81.5% | 52.3% | 53.1% |
> |      |            |------|------|------|------|------|------|------|------|------|------|
> | 1B | Annotated | 31.5% | 57.1% | 62.4% | 68.0% | 45.3% | 33.8% | 68.2% | 87.8% | 55.0% | 56.6% |
> | 1B | SFT | 30.2% | 59.6% | 61.7% | 76.0% | 46.5% | 33.4% | 69.1% | 88.7% | 54.5% | 57.7% |
> |      |            |------|------|------|------|------|------|------|------|------|------|
> | 2.5B | Annotated | 36.1% | 65.8% | 65.5% | 78.0% | 55.0% | 36.4% | 70.8% | 92.7% | 59.9% | 62.3% |
> | 2.5B | SFT | 37.7% | 67.1% | 57.2% | 81.0% | 55.8% | 37.0% | 72.3% | 92.6% | 61.0% | 62.4% |
>
> Performance is essentially unchanged: averages are 54.5/53.1 at 0.6B, 56.6/57.7 at 1B, and 62.3/62.4 at 2.5B for ours/baseline.
>
> > Do you believe this "inverse scaling" phenomenon will naturally disappear, or is it an inherent flaw of the standard SFT objective itself?
>
> Our framework suggests yes: if post-training data had sufficiently high semantic diversity, collapse should diminish. But that would partly defeat the purpose of pre-training + SFT: pre-training is valuable precisely because it provides massive, naturally diverse data, while SFT datasets are typically much smaller and more curated. We therefore view it as useful to have a post-training objective that preserves pre-training semantic diversity without requiring an SFT corpus of comparable scale. This also matters in limited-data settings such as personalization [1] or test-time training [2], where collapse can still arise.
>
> —
>
> In summary, we added evidence on evaluator independence and capability retention, clarified the compute tradeoff, and will strengthen the limitations discussion. We hope this clarifies the scope and relevance of our contribution, and we would appreciate your consideration in light of these additions.
>
> Thank you very much,
>
> Authors
>
> [1] Woźniak et al., Personalized Large Language Models (2024)
>
> [2] Hardt & Sun, Test-Time Training on Nearest Neighbors for Large Language Models (2023)

---

> > ### Author Rebuttal · Reviewer_XbLF · 2026-04-03
> >
> > Thanks for your detailed response, which addressed my major concerns #2 and #3 (but #1 remained). I will update my score to 4. Good luck!

---

### Official Review · Reviewer_sAmS · 2026-03-13

**Soundness:** 2
**Presentation:** 3
**Significance:** 2
**Originality:** 3
**Overall Recommendation:** 3
**Confidence:** 4

**Summary:**

Post-training via SFT improves instruction-following but collapses semantic diversity, and this worsens with scale. The paper proposes annotation-anchored training: during pre-training, documents are paired with semantic tags ("topic:Baghdad", "entity:Mir Hussein"); during post-training, annotation tokens are masked from the loss, freezing the annotation distribution while updating response behavior; at inference, the model samples an annotation first, then generates a response. This preserves pre-training semantic diversity while adopting SFT improvements reporting 6x less diversity collapse than standard SFT at 2.5B parameters.

**Compliance With Llm Reviewing Policy:**

Affirmed.

**Key Questions For Authors:**

- Why are annotations necessary over simpler alternatives like KL regularization toward the base model or inference-time base model steering? Without these baselines the necessity of annotated pre-training is not established.
- Annotation tokens replace content tokens within a fixed budget. Have the authors measured downstream capability (MMLU, coding) to confirm this does not degrade knowledge or reasoning?
- The quality evaluator is Qwen3-30B, the same family as the annotation model. Have the authors tested with an independent judge?
- What fraction of meaningful semantic variation falls outside the annotation schema? If diversity gains are bounded by annotation entropy, how much of the full semantic space does the schema cover?

**Limitations:**

The limitations section does not address the method's inapplicability to existing models, the lossy compression of semantic space, the potential capability cost of replacing content tokens, or the absence of simpler baselines

**Strengths And Weaknesses:**

Strengths:
- The core insight is clean: SFT entangles semantic intent and response quality in a single loss, annotations allow you to mask one while updating the other
- The factorized view in Equations 1-3 makes the theoretical motivation precise
- The SimpleStories controlled study is convincing: annotation-anchored models are essentially invariant to post-training data entropy, while standard SFT tracks it closely
- Ablations are well-designed and rule out simpler explanations: both annotated pre-training and inference-time annotation sampling are shown to be necessary
Weaknesses:
- The method requires pre-training from scratch and cannot be applied to any existing model. At frontier scale the annotation pipeline (3 days on 8xH100 for 50B tokens) would take months for a trillion-token corpus
- No comparison against simpler alternatives that also target the semantic marginal: KL regularization toward the base model, representation freezing, or inference-time base model steering (Wang et al., 2025) are all plausible approaches requiring no annotated pre-training
- The annotation schema is a lossy compression of semantic space. Diversity gains are upper-bounded by annotation entropy, and semantic variation outside the schema cannot be recovered
- Annotation tokens replace content tokens within a fixed token budget. The model sees fewer raw content tokens, which could hurt downstream knowledge and reasoning. This is never measured and quality is evaluated only via LLM judge on creative tasks
- The quality evaluator is Qwen3-30B, the same family as the annotation model, introducing potential bias
- Positive scaling is extrapolated, not demonstrated as models are only trained up to 2.5B parameters

---

> ### Author Rebuttal · Authors · 2026-03-31
>
> Dear Reviewer,
>
> Thank you for your thoughtful feedback. We respond inline below.
>
> > The method requires pre-training from scratch and cannot be applied to any existing model. At frontier scale the annotation pipeline (3 days on 8xH100 for 50B tokens) would take months for a trillion-token corpus
>
> This is a fair concern. However, our goal is to understand how to intervene during pre-training to produce models that can later be post-trained without semantic mode collapse. We believe this is broadly relevant to the ML community, since recent language models still exhibit substantial semantic collapse after post-training (Figure 2), so this remains an important open problem.
>
> > No comparison against simpler alternatives that also target the semantic marginal
>
> We have added an experiment comparing against SFT + KL regularization toward the base model, with regularization strength λ. We evaluate on Stories using the same setup as in the paper (analogous to Figure 5):
>
> ||Annotated|KL (λ=0.01)|KL (λ=0.1)|KL (λ=1.0)|
> |-|-|-|-|-|
> |Diversity|8.282|6.643|6.969|8.035|
> |Quality|3.167|3.38|3.193|2.394|
> For λ = 0.01/0.1, diversity (6.64/6.97) is substantially below annotation-anchored training (8.28). For λ = 1.0, diversity remains lower (8.04), while quality drops substantially (2.39 vs. 3.17). Thus, annotation-anchored training provides a better diversity-quality tradeoff than KL regularization.
>
> This establishes that annotation-anchored training outperforms KL regularization on the diversity-quality tradeoff.
>
>
> > Diversity gains are upper-bounded by annotation entropy, and semantic variation outside the schema cannot be recovered
>
> We agree that the schema is lossy and cannot capture the full semantic space. To clarify the breadth of the annotation distribution, we measure the number of unique annotations: 152M unique values across 156M documents and 2.3B total annotation key-value pairs. This reflects a highly open-ended annotation prompt (Appendix B.2). Our claim is therefore not that annotations recover all semantic variation, but that preserving a coarse yet diverse semantic scaffold already substantially mitigates collapse.
>
> > The model sees fewer raw content tokens, which could hurt downstream knowledge and reasoning.
>
> We have added an evaluation on standard small-model LLM benchmarks (the same datasets reported, e.g., for OLMo-1B):
> |Size|Model Type|arc_challenge|arc_easy|boolq|copa|hellaswag|openbookqa|piqa|sciq|winogrande|Average|
> |-|-|-|-|-|-|-|-|-|-|-|-|
> |0.6B|Annotated|28.9%|53.1%|61.4%|73.0%|40.1%|30.8%|65.9%|85.0%|52.3%|54.5%|
> |0.6B|SFT|28.4%|53.6%|55.4%|68.0%|41.1%|31.4%|65.7%|81.5%|52.3%|53.1%|
> |||-|-|-|-|-|-|-|-|-|-|
> |1B|Annotated|31.5%|57.1%|62.4%|68.0%|45.3%|33.8%|68.2%|87.8%|55.0%|56.6%|
> |1B|SFT|30.2%|59.6%|61.7%|76.0%|46.5%|33.4%|69.1%|88.7%|54.5%|57.7%|
> |||-|-|-|-|-|-|-|-|-|-|
> |2.5B|Annotated|36.1%|65.8%|65.5%|78.0%|55.0%|36.4%|70.8%|92.7%|59.9%|62.3%|
> |2.5B|SFT|37.7%|67.1%|57.2%|81.0%|55.8%|37.0%|72.3%|92.6%|61.0%|62.4%|
>
> Overall, the annotated models have comparable average performance to standard SFT models: 54.5% vs. 53.1% at 0.6B, 56.6% vs. 57.7% at 1B, and 62.3% vs. 62.4% at 2.5B.
>
> > The quality evaluator is Qwen3-30B, the same family as the annotation model, introducing potential bias
>
> We have added an experiment using GPT-5.4 as an independent LLM judge for diversity evaluation. On Stories (analogous to Figure 5), we obtain:
> |model_type|temp=0.6|temp=0.9|temp=1.0|temp=1.05|temp=1.1|
> |-|-|-|-|-|-|
> |Vanilla (2.5B)|5.07|6.88|7.39|6.94|6.41|
> |Annotated (2.5B; ours)|*7.13*|*8.36*|*8.1*|*7.39*|*6.8*|
> Annotation-anchored training still shows a substantial diversity improvement over standard SFT under this independent evaluator. We will replace the paper’s evaluations with this GPT-5.4-based annotator.
>
> > Positive scaling is extrapolated, not demonstrated as models are only trained up to 2.5B parameters
>
> Compute limits prevent us from training beyond 2.5B. However, we do establish that the inverse scaling trend of diversity in standard post-training holds well beyond this range by evaluating Qwen and Llama models up to 70B. We also show across 0.6B-2.5B that our method restores positive scaling of diversity. While we cannot prove this trend continues indefinitely, we believe the result is still meaningful at this scale.
>
> —
>
> In summary, we have added experiments comparing against simpler baselines, evaluating general capability on standard benchmarks, quantifying annotation coverage, and using an independent judge. These address most of your concerns. For the remaining limitations regarding scale and applicability, we will clarify them in the limitations section. We hope these additions make clear that our methodology is relevant to the broader ML community interested in pre-training.
> We very much appreciate your review, and we would appreciate your considering a higher score in light of these new experiments.
>
> Thank you very much,
>
> Authors

---

> > ### Author Rebuttal · Reviewer_sAmS · 2026-04-04
> >
> > My concerns have been addressed but I am not fully convinced about the practicality of the method when it comes to existing models which explains my reluctance to raise my score.

---

### Decision · Program_Chairs · 2026-04-30

**Decision:**

Accept (regular)

**Comment:**

The paper focuses on the mode collapse problem in LLM post-training, i.e., supervised fine-tuning biases models toward low-entropy distributions and reduces output diversity, which worsens with scale. It proposes an annotation-anchored training method and augments pre-training with semantic tags and masks to preserve the model's pre-trained distribution diversity while learning response behavior. At inference time, the model first samples a semantic plan (tags) and then generates text conditioned on them.

The reviewers believe the paper is well motivated by an important problem, post-training collapse; the proposed method is intuitive and well-motivated; experiments are well-designed, and results are convincing.

After rebuttal, most reviewers have agreed that most concerns have been successfully addressed, such as i) the lack of evaluation on downstream task performance/capabilities (e.g., MMLU); ii) LLMs from the same family for both annotation and judgement; iii) no comparison with other baselines, e.g., KL regularization; and iv) influence on non-creative tasks. In the AC-Reviewer discussion phase, Reviewer r4Ha (with Recommendation=3) acknowledged the added GSM8K results are reassuring and believed that the contributions to open-ended generation are solid.

The two major remaining concerns are 1. The method applies to pretraining and thus requires substantial computational cost due to annotation construction and increased token usage. 2. Limited evidence on non-creative tasks (e.g., math reasoning).

For the first one, the authors have estimated annotation cost and demonstrated that annotating the corpus for a 70B model constitutes only 1.5% of the total training cost, thereby partially supporting the method's practicality. For the other, the authors' explanation appears reasonable, and their claims regarding creative tasks are well supported. Although closed-domain tasks are standard in LLM training, cross-domain generality, as well as training efficiency, are not claimed as central contributions by the authors.

Put all together, we believe the main claims of this work have been well supported. We highly encourage the authors to fully address all raised concerns and include all responses and additional results in the revision.

Besides, the following reference has been flagged by the ICML system as a hallucinated one. However, after checking it, I think this is a false positive. Please verify the reference and address any potential issues (e.g., formatting or citation inconsistencies) that may cause this flag:

Reference: Finke, L., Sreedhara, C., Dooms, T., Allen, M., Zhang, E., Rodriguez, J. D., Nabeshima, N., Marshall, T., and Braun, D. Parameterized synthetic text generation with simplestories. arXiv preprint arXiv:2504.09184, 2025.